# Evolving Altruistic Attitudes towards Vaccination Post COVID-19 Pandemic: A Comparative Analysis across Age Groups

**DOI:** 10.3390/vaccines12050454

**Published:** 2024-04-24

**Authors:** Verena Barbieri, Christian J. Wiedermann, Stefano Lombardo, Giuliano Piccoliori, Timon Gärtner, Adolf Engl

**Affiliations:** 1Institute of General Practice and Public Health, Claudiana College of Health Professions, 39100 Bolzano, BZ, Italygiuliano.piccoliori@am-mg.claudiana.bz.it (G.P.);; 2Department of Public Health, Medical Decision Making and Health Technology Assessment, University of Health Sciences, Medical Informatics and Technology, 6060 Hall, Tyrol, Austria; 3Provincial Institute for Statistics of the Autonomous Province of Bolzano—South Tyrol (ASTAT), 39100 Bolzano, BZ, Italy

**Keywords:** altruism, vaccine uptake, health behavior, COVID-19, ECRC altruism scale, IPIP 5F30F-R1 altruism scale, health policy

## Abstract

Altruism plays an essential role in promoting vaccine uptake, an issue that came to the fore during the COVID-19 pandemic through discussions of herd immunity and altruistic motivations. In response, the primary objective of this cross-sectional survey was to explore how altruistic attitudes have evolved in the post-pandemic era and to assess their effectiveness in motivating vaccination behavior in different age groups. The study aimed to elucidate changes in altruistic motivations for vaccination and their implications for public health strategies. Using a representative sample of the adult population of South Tyrol, Italy, including 1388 participants, altruism was assessed in 2023 with the scales of the Elderly Care Research Center (ECRC) and the International Personality Item Pool (IPIP) subscale of the version 5F30F-R1. Its association with demographic variables, vaccination attitudes and personal beliefs in two age groups (18–69 years, 70+ years) was analyzed. The results reveal distinct predictors of altruism across these scales and age groups, suggesting a shift in altruistic attitudes towards vaccination when comparing data from a similar survey conducted in 2021 with the 2023 results. Consequently, the use of altruism scales for different age groups is warranted. This study highlights the need for further research in this field. It concludes that while promoting altruistic behavior to increase vaccine uptake appears to be effective primarily among the younger population, emphasizing personal safety is more appropriate for encouraging vaccination among older individuals.

## 1. Introduction

Vaccination is a major medical advance, with uptake being influenced by perceived risk, confidence in vaccine safety, and social norms [1]. In particular, altruism plays an influential role in vaccination decisions [2], with targeted pro-vaccine messages increasing vaccination intentions [3]. Concern for community well-being motivates participation in herd immunity efforts [4], suggesting that emphasizing the community benefits of vaccination can strengthen public health campaigns. Altruism has also shifted influenza vaccination decisions towards prioritizing community health [5] and improved attitudes towards childhood influenza vaccination among hesitant parents [6].

Altruism is defined as the practice of selfless concern for the well-being of others without expecting immediate personal gain, although it ultimately benefits the actor more than the costs incurred. This voluntary behavior demonstrates consideration for others motivated by intrinsic factors rather than immediate rewards [7,8]. Research has consistently demonstrated that altruism and associated behaviors such as empathy and prosocial actions tend to increase with age. Older adults display a greater propensity for altruistic behavior than their younger counterparts [9,10]. This inclination towards empathy and prosocial behavior is more pronounced in the elderly [11]. Moreover, a meta-analysis underscores a direct correlation between age and altruism, noting that diminishing availability of personal resources in older adults has a moderating influence on this relationship [12]. Interestingly, while higher levels of altruism are observed in older individuals, the motivations driving altruistic behavior differ by age group, with younger individuals often experiencing stress [13]. This distinction suggests that the underlying motivations for altruism evolve over a lifespan, reflecting changes in life circumstances and psychological priorities.

Given the evidence that altruism is influenced by different factors in older individuals, the Elderly Care Research Center (ECRC) altruism scale has emerged as a significant tool for evaluating altruism in those aged 70 and above [14]. Although this questionnaire has been validated for use across various age groups, its primary application has been for older demographics. This raises an important consideration of the appropriateness and effectiveness of utilizing such specific altruism scales for studying altruistic behavior across the entirety of an aging population [15]. A contrastive strategy may be necessary to fully capture the complexity of altruism across age groups. Dividing the population into two age groups (18–69 and 70+) may allow for a more focused analysis of age-related altruistic influences. In addition, using different measures for each group may better capture the different drivers and manifestations of altruism.

Age was positively correlated with enhanced prosocial behavior during the coronavirus disease 2019 (COVID-19) pandemic [16], and recent findings indicate that altruism and accumulated life experience of older adults significantly contributed to the promotion of community health during this period [17]. During the pandemic, altruism significantly influenced health behaviors and vaccine uptake, with a notable correlation between age and prosocial behavior [18]. Altruistic motives are key to the development and uptake of COVID-19 vaccines [19], supported by the association between organ transplantation rates and vaccine uptake in the European Union [20,21]. This suggests the effectiveness of altruistic appeals in vaccination campaigns, as observed in the United Kingdom [22]. In Germany, a study found associations between altruism and factors such as gender, parenthood, physical activity, and health status in adults aged 18–69 years [23]. These findings suggest integrating altruism into health behavior research and public health strategies, emphasizing tailored approaches for different age groups to strengthen vaccination efforts.

In a 2021 study conducted in South Tyrol [24,25], attitudes towards COVID-19 vaccination were assessed in the age groups 18–69 and 70+ years [15] using the ECRC altruism scale [14]. In line with the findings of Hajek and König [23], the analysis differentiated the impact of the identified predictors on these attitudes. The results indicated that the older demographic group exhibited higher altruism and a stronger inclination towards both COVID-19 and general vaccination policies. In addition, a notable shift from altruistic to self-interested motivations for adherence to pandemic guidelines was observed in the older age group [15]. These findings highlight the importance of altruism in the development of health communication and vaccination strategies and reflect different age-related responses to pandemic measures.

This study aims to elucidate the role of altruism in shaping vaccination attitudes in the post-pandemic era by exploring the nuanced relationship between altruistic motivations and vaccination preferences: (i) It critically examines whether differences in altruistic attitudes towards COVID-19, general, and influenza vaccinations emerge in the post-pandemic period compared to the pandemic period. (ii) Given the variability in altruistic behavior across age demographics and the existence of age-specific altruism scales, such as the ECRC scale for the elderly, this study questions the efficacy of using different scales to assess altruism across age groups. (iii) It also examines potential predictors of altruism, including demographic factors, personal beliefs, and attitudes towards vaccination and how these variables may differ across age groups. (iv) Finally, this study seeks to identify methods to motivate individuals of different ages to vaccinate, particularly in the context of mandatory vaccination policies, thus contributing to the broader discourse on increasing vaccine uptake through altruistic appeals within public health strategies.

## 2. Materials and Methods

Consensus-based checklist recommendations for reporting survey studies (CROSS) were followed [26].

### 2.1. Study Design and Data Collection

Data collection was facilitated by a modified COSMO survey [27] conducted in South Tyrol in February 2023, targeting individuals over the age of 17 years using a probability-based sampling method. The clarity and reliability of the COSMO questionnaire were assessed using an established survey evaluation tool [28]. Full details of the study methodology, including participant recruitment and sample size determination, have been previously described [29]. South Tyrol (Province of Bolzano), the northernmost region of Italy, has a population of approximately 530,000.

### 2.2. Altruism

Altruism was measured using two scales. The ECRC scale [14], known for its brevity, reliability, and validity, has been used to assess prosocial orientation, particularly in individuals aged 70 years and older, although its application extends to other age groups. Participants responded to items on a 6-point Likert scale ranging from 1 (“strongly disagree”) to 6 (“strongly agree”), including “I enjoy doing things for others”, “I try to help others even if they don’t help me”, “Seeing others succeed makes me happy”, “I really care about other people’s needs”, and “I come first and shouldn’t have to worry so much about others”. This scale allowed for a nuanced assessment of altruistic tendencies within the study population. Strong correlations between the altruism scale, salient personality traits, psychological well-being, religiosity, and meaning in life establish the construct validity. The items are summed to a total score, which has been well researched and ranges from 5 to 30, with 5 representing the lowest and 30 representing the highest level of altruism [14].

Second, altruism was measured using the ‘altruism’ subscale of the version 5F30F-R1 [30] from the International Personality Item Pool (IPIP) [31], specifically its validated German version [32]. This subscale comprises six items rated on a five-item Likert scale from “strongly disagree” to “strongly agree”. The questions are made of the statements, (IPIP_1) “The needs of others for me are ranked first”, (IPIP_2) “I’m more concerned about others than about me”, (IPIP_3) “I have a good word for everyone”, (IPIP_4) “I back up others”, (IPIP_5) “I anticipate the needs of others to make them feel good” and (IPIP_6) “I make people feel welcome”. The items were summed to obtain a total score that was examined in detail and could take values from 6 to 30, with 6 indicating the lowest and 30 indicating the highest level of altruism [30].

### 2.3. Questionnaire

Sociodemographic data were collected, asking about age, gender, economic situation in the last three months (better, same, worse, do not know), educational level (primary school, vocational school, high school, university), Italian nationality, having a chronic disease, living with vulnerable people, and community.

Other items included trust in the national vaccination plan, COVID-19 vaccination coverage, trust in institutions (health authorities and politics) [33,34], conspiracy thinking [35], well-being [36], spirituality [37], and consultation with complementary and alternative medicine (CAM) providers over the past 12 months [38,39].

Trust in institutions was measured on a 6-point Likert scale from “no trust” to “great trust” and a seventh item “don’t know”. Conspiracy perceptions (5 questions on a 6-point Likert scale from “strongly disagree” to “strongly agree”), spirituality (7 questions on a 4-point Likert scale from “never” to “regularly/very often”), and well-being within the past two weeks (5 items on a 4-point Likert scale from “all the time” to “never”). The summed scores of these variables were considered potential predictors of altruism after the COVID-19 pandemic. Agreement with decisions made during the pandemic regarding restrictions and vaccination was assessed using individual questions [40]. Agreement was asked on a 6-point Likert scale from 1 (strongly disagree) to 6 (strongly agree) and a seventh item “I don’t know what they decided”. Questions about trust in vaccination and COVID-19 vaccination were asked on a 6-point Likert scale ranging from 1 (strongly disagree) to 6 (strongly agree), as previously described [24]. The question “I had the opportunity to discuss the COVID-19 vaccination with a doctor or health care worker from the health district” was answered dichotomously by yes or no.

### 2.4. Altruism Scales and Age Groups

The ECRC altruism scale was developed specifically for adults over the age of 70, but can also be used in younger age groups [14]. The present study examined the effects of the predictors of altruism identified by Hajek and König [23], who used the IPIP altruism scale [30], on COVID-19 vaccination and attitude. While Hajek and König only used the IPIP altruism scale, our study compares ECRC and IPIP altruism scales. Since the ECRC scale has been validated for people aged 70 years and older, and Hajek and König referred to people 18–70 years of age, the younger age group of 18–69 years and the age group of 70 years and older were separated for the present analyses. Using both the ECRC and IPIP altruism scales in the same survey allows cross-validation to enhance the robustness and applicability of the findings of altruistic behavior predictors across different age groups. Furthermore, it will be possible to highlight differences between younger (mostly working) and older (mostly retired) participants, who experience life and especially the time of the pandemic, from a different point of view.

### 2.5. Statistical Analysis

Cronbach’s alpha was calculated for all the sum scores to test reliability. Metric data were not normally distributed and are presented as medians, interquartile ranges, and mean ± standard deviation (SD). Significant differences between the groups were tested using the Mann–Whitney U test. Nominal and ordinal data are presented as absolute numbers and percentages. The chi-square test was used to test for differences, and Kendall’s tau was used to test for correlations with dichotomous variables. All other correlations were calculated using Spearman’s correlation coefficients. Given the exploratory nature of the study and the focused number of comparisons, alpha (α) adjustment for multiple comparisons was not performed.

The sum scores were calculated for the ECRC and IPIP scales of altruism, conspiracy theories, well-being, spirituality, and trust in institutions, as described above. For all Likert-scale questions with an additional “I don’t know” or “I don’t know the decision” item, this item was replaced by the mean of the scale in order to use the entire dataset for the sum score calculations. Altruism scores were normalized using the formula (x − x_min_)/(x_max_ − x_min_), where max and min are the absolute minimum and maximum possible values of the sum scores, respectively.

Linear multiple regression models were used to fit altruism based on the predictor variables for both age groups. Initial candidate predictors were identified based on their theoretical relevance to altruism and vaccination attitudes, as established by existing literature. This was followed by preliminary bivariate analyses to assess the relationship between each candidate predictor and the altruism scales. Only variables demonstrating significant associations in these preliminary analyses were considered for inclusion in the final models. For a linear regression model with 13 predictors, a minimum sample size of 251 was required, assuming a type 1 error of 5%, a power of 95%, and a small effect size of 0.11 (corresponding to a squared R of 0.1, which is significantly different from 0). For the older age group, where only a few variables were included in the model, a sample size of 172 was sufficient when four variables were used. The sample size calculation was performed using G*Power version 3.1. Regression was checked for linear relationships between predictors and independent variables, and regression diagnostics were performed by checking for normality and mean 0 of residuals, homoscedasticity, multicollinearity, autocorrelation of error terms using the Durbin–Watson test, and outliers using the DF-beta statistic, Cook distance, and leverage diagnostics.

*p*-values < 0.001 are indicated with ***, <0.01 with **, <0.05, *, and *p*-values ≥ 0.05 are considered not significant (n.s.). All statistical analyses were performed using the IBM SPSS Statistics for Windows (version 27.0; IBM Corp., Armonk, NY, USA).

## 3. Results

### 3.1. General Characteristics of the Dataset and the Two Altruism Scales by Age Group

Data were collected from 1388 individuals. The group of participants aged 18–69 years included 1184 individuals and 204 participants aged ≥ 70 years. The demographic characteristics of the dataset were representative of age, gender, and community.

In assessing the reliability of the instruments within the two age groups, the following Cronbach’s alpha values were used for the ECRC and IPIP altruism scales: 0.77 and 0.80, respectively, for the ECRC altruism score in the younger and older age groups; and 0.87 and 0.90, respectively, for the IPIP altruism score. In addition, separate instruments were used to assess other constructs of interest. The reliability of these instruments, as indicated by Cronbach’s alpha values, was as follows: for spirituality, 0.87 in the younger age group and 0.85 in the older age group; for well-being, 0.85 and 0.86; for conspiracy thinking, 0.84 and 0.81; and for trust in institutions, 0.91 and 0.90.

The ECRC scale total scores ranged from 5 to 30 and the IPIP scale total scores from 6 to 30, with a median altruism total score of 23 (interquartile range [Q1;Q3] = [19;26]) for ECRC and 20 (interquartile range [17;24]) for IPIP. The normalized scores reached an overall mean (SD) of 0.59 (0.219) for ECRC and 0.70 (0.187) for IPIP. Spearman’s correlation coefficient for the correlation between age and the two altruism scores was not significant in either case. No significant difference in altruism scores was found between the two age groups.

Figure 1 presents the normalized mean scores from the ECRC and IPIP altruism questionnaires for various age groups. The ECRC scores were consistently higher across the board. Additionally, the right panel illustrates a modest correlation between age and the difference in scores (normalized ECRC minus normalized IPIP) between the two questionnaires, with Spearman’s rho calculated at 0.076, indicating a slight but statistically significant relationship (*p* < 0.01). This suggests that age may play a role in altruistic self-perception, as measured by these scales, warranting further examination of the individual items in each questionnaire.

Figure 2 displays the participant agreement percentages with items from the ECRC and IPIP altruism scales across the two age groups. Significant differences between age groups were observed for ECRC–3 (*p* = 0.023), IPIP–4 (*p* < 0.001), and IPIP–6 (*p* = 0.021) using the Mann–Whitney U test. Kendall’s tau-b indicated significant associations for ECRC–3 (0.051, *p* < 0.05, more agreement in the older age group), IPIP–4 (−0.093, *p* < 0.01, less agreement in the older age group), and IPIP–6 (−0.053, *p* < 0.05, less agreement in the older age group).

Spearman’s rank correlation coefficient revealed significant relationships between age and agreement levels for certain items. ECRC–3 showed a positive correlation (0.057, *p* < 0.05), indicating higher agreement in older participants, whereas ECRC–5 (−0.081, *p* < 0.01), IPI–1 (0.069, *p* < 0.01), and IPIP–3 (−0.062, *p* < 0.05) displayed varying correlations. Notably, IPIP–4 (−0.179, *p* < 0.001) and IPIP–6 (−0.049, *p* < 0.05) were negatively correlated, suggesting a lower agreement with increasing age.

### 3.2. Demographic and Attitudinal Profiles by Age Group

The demographic and health-related variables revealed distinct patterns across the two age groups (Table 1). Gender distribution did not significantly differ between the two age groups, nor did discussions about COVID-19 vaccination with healthcare professionals. Older participants (70+ years) were notably more likely to be Italian nationals, have chronic diseases, reside in urban areas, experience stable economic situations, agree with national vaccination plans, receive flu vaccinations, have multiple COVID-19 vaccinations, and support childhood vaccinations for herd immunity. Younger adults (18–69 years) had higher levels of education and more frequent use of complementary and alternative medicine (CAM) within the past year. The sum scores of personal beliefs differed significantly between age groups for trust in institutions (18–69: median [IQR] = 29 [23;35]; 70+: 30.5 [24;37]; *p* = 0.04), but not for spirituality, conspiracy thinking, and well-being.

### 3.3. Altruistic Behaviors and Vaccination Attitudes in Diverse Age Populations

#### 3.3.1. Associations between Altruism, Population Characteristics and Vaccination-Related Actions

Table 2 details the relationships between altruism scores and a range of demographic and attitudinal variables for different age groups, using the ECRC and IPIP scales. Consistently across age groups, a significant positive correlation exists between the ECRC and IPIP altruism scores, demonstrating the scales’ coherence in measuring altruistic behavior. Notable findings include the positive association of female gender with altruism scores in both age cohorts and the link between opportunities for COVID-19 vaccination dialogue and higher altruism scores. Among younger participants, factors such as trust in the national vaccination plan, belief in herd immunity, and COVID-19 vaccine uptake correlate positively with altruism scores. The IPIP score, particularly in the younger group, also shows a positive relationship with urban residency and living with at-risk individuals. While spirituality and trust in institutions are positively associated with altruism across all participants, notable differences in other variables such as education level and CAM use are observed between age groups.

The analyses also focused on the relationship between altruistic behavior and vaccination practices, including general attitudes towards vaccination, COVID-19 vaccine receipt, influenza vaccination history, and engagement with CAM providers. Significant associations were found among these factors only within the 18–69 age group. The ability to discuss COVID-19 vaccination with healthcare professionals was positively associated with altruism in both the age groups. For the 18–69 age group, the box plots in Figure 3 illustrate higher altruism scores among younger participants who agreed with national vaccination plans or had received multiple COVID-19 vaccinations. Younger individuals who had consulted a CAM provider within the past year showed greater altruism on the ECRC scale, an effect that was not reflected in IPIP scores. No significant effects were found in the older age group; thus, box plots are not presented.

In the younger age group, participants who received the flu vaccine exhibited significantly lower altruism levels according to the ECRC scale (*p* < 0.001), a trend that was not observed with the IPIP scale (*p* ≥ 0.05), nor on either scale in the older age group.

Individuals who reported having had the opportunity to discuss vaccination with healthcare professionals demonstrated significantly higher altruism levels, as measured by the ECRC scale in both age groups (*p* = 0.043 and *p* = 0.04, respectively) and the IPIP scale in the younger age group (*p* < 0.001). This association suggests that the ability to engage in informed conversations regarding vaccination may reflect broader altruistic tendencies.

#### 3.3.2. Altruism and Vaccination Perspectives

Table 3 explores how altruistic tendencies, as measured by the ECRC and IPIP scales, correlate with attitudes towards vaccination policies, trust in COVID-19 vaccination, and skepticism about vaccine risks across different age groups. Key observations include a positive alignment of younger individuals’ altruistic scores with supportive attitudes towards political decisions on COVID-19 and mandatory vaccinations, and beliefs in the efficacy of COVID-19 vaccines to contain the virus’s spread. Interestingly, long-term vaccine risks did not significantly correlate with altruism, highlighting nuanced perceptions of vaccine safety and efficacy among participants. Additionally, skepticism expressed by medical professionals showed a unique positive correlation with the IPIP score among younger participants, underscoring the influence of professional opinions on altruistic vaccination attitudes. Negative correlations were observed in younger participants concerning mandatory vaccination’s necessity, indicating varied stances on vaccination policies. Overall, the table sheds light on the complex relationship between altruism and vaccination attitudes, suggesting that tailored communication strategies might be more effective in addressing vaccine hesitancy.

### 3.4. Regression Analyses

Table 4 outlines the predictors of altruism across age groups, determined through linear regression analyses that control for a comprehensive range of variables, highlighted by significant findings from Table 3. Key determinants include gender, trust in institutions, and spirituality, with unique factors such as agreement on herd immunity’s importance also playing a role, particularly in the younger cohort.

The regression models’ integrity was rigorously tested: the VIF values were below 1.2 across all models, indicating no concerns of multicollinearity. The Durbin Watson Statistic ranged from 1.648 to 2.105, suggesting that autocorrelation did not compromise the analyses. However, while DF-beta statistics and Cook’s distance analyses identified outliers, their exclusion had minimal impact on the models’ primary predictors, albeit revealing some additional weakly significant factors in the younger cohort. This robustness check, reflected in the corrected R^2 values, underscores the models’ validity in capturing the nuances of altruistic behavior comparing the age groups 70+ and 18–69 years, with explained variances ranging from 0.107 to 0.203.

These diagnostics confirm the models’ reliability in exploring altruism’s determinants, emphasizing the influence of communicative opportunities on vaccination attitudes among the younger participants. The detailed statistical analysis, including the management of outliers and checks for multicollinearity and autocorrelation, ensures the conclusions drawn are both robust and insightful, as further detailed in Table 4.

## 4. Discussion

In the aftermath of the COVID-19 pandemic, this study examined the complex relationship between altruism and vaccination attitudes. At the core of the investigation is a critical examination of whether differences in altruistic attitudes towards COVID-19, general, and influenza vaccination evolved in the post-pandemic era. To enhance the assessment of altruism, this study examines the effectiveness of using two different altruism scales, the ECRC scale for older populations and the IPIP scale, to assess altruistic behaviors comparing the age groups 70+ and 18–69 years. Through rigorous analysis, this research identifies key demographics, personal beliefs, and attitudinal predictors of altruism and how these elements interact across age groups. The results suggest that while general altruism and prosocial behavior typically increase with age, vaccination decisions in the post-pandemic period show a differentiated divergence. Older adults may prioritize personal safety over altruistic motives due to higher health risks, whereas younger individuals may be more influenced by altruistic reasons, possibly driven by a sense of community responsibility. This highlights the need for tailored public health messages that take into account the different motivations of different age groups to increase vaccine uptake.

### 4.1. Post-Pandemic Shifts in Altruism and Vaccination Attitudes

This study aimed to explore altruism within the context of vaccination among adults in South Tyrol, Italy, following the pandemic. Mirroring our investigation during the pandemic [24,25], we previously identified a significantly higher level of altruism in the 70+ age group (median ECRC score = 24) than in the 18–69 years age group (median = 23). Our current findings, however, indicate that this age-related difference in altruism levels has equalized (median = 23 for both groups), with no significant differences in IPIP altruism scores between the two age groups. Earlier research corroborates the notion of age-related altruism [9,10,11,12,13] and a heightened sense of altruism towards those in closer social proximity [16,17]. Notably, in 2021, a higher ECRC altruism score was significantly linked not only to age but also to cohabitation with COVID-19 patients at risk [15]. In contrast, this 2023 analysis did not reveal a significant correlation with age and close relatives, although the IPIP altruism score showed a significant correlation with living with COVID-19 patients at risk exclusively in the older demographic group. Variations in agreement with the individual altruism scale statements between age groups were observed.

The affirmation ‘Seeing others prosper makes me happy’ (ECRC-3) received more consensus among the older population, whereas ‘I back up others’ (IPIP-4) and ‘I make people feel welcome’ (IPIP-6) were more prevalent in the younger demographic. This diverges from 2021, when more ECRC statements garnered significant agreement in the older age group, raising the question of whether altruistic attitudes, especially among older individuals, have waned during the pandemic [15].

Regarding vaccination attitudes, the 2021 analysis revealed a significant association between a positive stance on COVID-19 vaccination and higher ECRC altruism scores in both age groups [15], a connection that has weakened in 2023. This observation extends to perceptions of decision-making efficacy and the perceived necessity and safety of COVID-19 vaccines across both groups, with IPIP altruism scores showing faint associations with these attitudes.

In 2021, skepticism towards childhood vaccination’s necessity and safety was not linked to the ECRC altruism score in either age group [15]. Intriguingly, in 2023, such attitudes are negatively correlated with the ECRC altruism score in the younger demographic but not with the IPIP score. Minimal associations were noted in the older group for both the scores. This prompts consideration of the appropriateness of the ECRC scale for younger or working individuals and suggests that vaccination beliefs might not be directly tied to altruism, despite proposals for altruism-based strategies to mitigate vaccine hesitancy [41].

The analysis underlines the positive correlation between altruism and the importance attributed to childhood vaccination for child protection and herd immunity across both age groups and scores. Concerns about the pandemic’s impact on vaccination rates were relevant to altruism, primarily in the younger cohort. Unlike in 2021, where no association was detected with the ECRC altruism score for either age group across all statements [15], the current data illustrate a shift from COVID-19 vaccination being primarily associated with altruism in 2021 to a stronger correlation with attitudes towards mandatory childhood vaccination in 2023. This evolution emphasizes the adaptability of altruistic behavior in response to prevailing challenges and underscores the need for further exploration of how the pandemic reshaped altruistic sentiments within the community.

### 4.2. Applicability of ECRC and IPIP Altruism Scales across Age Groups in Vaccination Contexts

The suitability of the ECRC altruism score for younger populations remains unclear. To address this, we examined the differences between ECRC and IPIP altruism scores in detail.

A comparison of the two altruism scores revealed that the median normalized agreement was lower for IPIP than for ECRC across both age groups. On examining individual statements, we observed that ECRC’s statements were broader in nature, whereas IPIP’s statements prompted direct comparisons with others. Overall, participants showed a greater tendency to agree with ECRC statements than with IPIP statements, with younger individuals displaying a preference for IPIP statements and older individuals for ECRC statements. This trend suggests that the ECRC score may not fully capture the altruism of the younger working population given its validation primarily for older individuals. However, further analysis is required to gain a better understanding.

Significant differences in demographic characteristics between the two age groups were observed, underlining the rationale for separate analyses. Specifically, the ECRC’s validation for individuals over 70, coupled with changing motivations for altruistic behavior in older individuals [12,13], supports this differentiation. Moreover, vaccination attitudes varied significantly between age groups, with younger individuals showing less favor towards all types of vaccinations—COVID-19, influenza, and general childhood vaccinations. This aligns with previous findings regarding COVID-19 and influenza vaccine uptake in Italy [24,25].

For all vaccination types, younger individuals who were supportive of vaccination exhibited higher altruism scores for both measures. Conversely, no significant correlation was found between altruism and vaccination attitudes in the older group.

### 4.3. Altruism Predictors Post Pandemic across Age Groups

During the pandemic, Hajek and König [23] identified positive correlations between the IPIP altruism score and factors such as female gender, younger age, chronic diseases, support for COVID-19 vaccination, and having children within the 18–69 age group. In 2023, these correlations could be reaffirmed for female gender and COVID-19 vaccine uptake, but not for the remaining factors.

Additionally, in both age groups, both IPIP and ECRC altruism scores were positively associated with trust in institutions and spirituality. A novel finding of the ECRC score in the older age group was positively correlated with the opportunity to discuss COVID-19 vaccination. Regression analyses further solidified trust in institutions and spirituality as predictors of altruism for both scores, and gender also influenced ECRC altruism scores. These models did not completely remain robust in the face of outliers. For the younger demographic, spirituality and trust in institutions emerged as significant predictors across both altruism measures, alongside gender and endorsement of the significance of herd immunity. The IPIP score reflects various demographic factors. However, both the models exhibited sensitivity to outliers.

This supports earlier findings on spirituality [41,42] and institutional trust [43] as consistent predictors of altruism, alongside gender, for the 18–69 age bracket [18]. However, caution is advised when interpreting these regression models because of their sensitivity to outliers and selective prediction capacity for each altruism score. Notably, for both scores, alignments with the national vaccination plan, COVID-19 vaccine acceptance, herd immunity acknowledgment, and discussions around COVID-19 vaccination were correlated positively in the younger age group. Conversely, many demographic and personal predictors were associated with only one of these two scores.

Therefore, it appears that for younger individuals, associations with altruism are contingent on the score used. Given that the IPIP score relates to a broader range of demographic factors, while the ECRC score aligns more closely with personal factors, it prompts reconsideration of the appropriateness of the ECRC score for younger groups. It might serve more as a reflection of personal beliefs than as a pure measure of altruism. If the IPIP score is considered a valid tool, clear demographic associations emerge in the younger group, a pattern not observed in the older demographic group.

The analysis fails to replicate the 2021 findings, where cohabitation with non-risk adults negatively predicted altruism. Neither residing with children nor with COVID-19 risk individuals emerged as definitive predictors, diverging from Jones et al. [22], who noted stronger altruism connections with close relatives over other vulnerable groups. This discrepancy raises questions about potential shifts in altruistic attitudes after the pandemic.

### 4.4. Altruism-Based Strategies for Vaccine Uptake across Age Groups

In our 2021 study, we observed a shift from altruistic vaccination attitudes to self-focused motives with increasing age [15], highlighting the role of prosocial daily behavior in fostering altruism [44]. The present survey noted that the older demographic, predominantly retired, experienced distinct needs and responses during the COVID-19 pandemic compared with their younger counterparts. Utilizing two measures of altruism, we found notable vaccination behavior patterns, primarily within the younger age group.

An intervention deploying videos with prosocial and altruistic messages was found to significantly enhance COVID-19 vaccination intentions, more so among younger participants than older ones [45]. This suggests that age-specific interventions could effectively bolster altruism and consequently vaccination willingness. Consistent with this, in the present study, the younger age group alone showed a correlation between the perceived importance of herd immunity and altruistic attitudes, underscoring the recent discussions on this topic [4,6,21,46].

Leveraging altruism in health communication strategies to promote vaccine uptake is a prevalent approach. Given our findings, we advocate the development of these strategies, with a particular emphasis on the younger working population. For older retired individuals, direct messaging of personal health benefits might be more effective in vaccination discussions.

Moreover, fostering trust in institutions has emerged as an age-transcendent strategy to enhance vaccination willingness. This approach could bridge the gap in vaccine uptake among different age groups. Although our study did not directly examine the influence of anti-vaccination statements in the media and social media, the role of these platforms in shaping public perceptions and trust in vaccination cannot be underestimated. During and after the SARS-CoV-2 pandemic, anti-vaccination groups effectively used these channels to sow doubt and undermine trust in health authorities and vaccination campaigns. Addressing misinformation and strengthening the scientific basis for vaccination decisions are critical steps in rebuilding public trust and ensuring the success of immunization programs.

Finally, the association of the IPIP altruism score with Italian nationality and rural/urban residency in the younger demographic underscores the potential of altruism-focused strategies tailored to diverse population subgroups. This differentiated approach could maximize the reach and effectiveness of vaccination campaigns, aiming for comprehensive coverage of the entire population.

The ethical interplay between individual autonomy and collective health benefits in vaccination underscores the critical debate in medical ethics [47]. Vaccinations present a dilemma: balancing personal choices against public health needs. Emphasizing informed, voluntary decisions in vaccination policies is essential to respect autonomy while recognizing the communal value of vaccinations for disease prevention. In the post-pandemic landscape, exploring the influence of altruism on vaccination attitudes is crucial for aligning individual rights with public health goals.

### 4.5. Strengths and Limitations

In South Tyrol, vaccination rates for COVID-19, influenza, and other mandatory vaccinations remained among the lowest in Italy. This study, pioneering in its focus on altruism in the South Tyrolean context in both 2021 [15] and 2023, represents the first attempt to analyze such behaviors in this region. However, the lack of established reference values for altruism limits the ability to make direct comparisons with other Italian regions and international counterparts. Furthermore, we acknowledge the potential oversight of not including all possible predictors of altruism, such as political orientation or physical activity, which could further elucidate the complex dynamics that influence altruistic behaviors.

The aim of this study was to explore changes in attitudes towards vaccination across different age cohorts and to assess the appropriateness of different tools for measuring altruism within these groups. The use of altruism measurement tools and their recommended applicability across age cohorts is based on a study conducted within the specific demographic and cultural context of South Tyrol, Italy. Given the localized nature of the research setting and the size of the elderly study sample, caution should be exercised in extending these findings to broader populations. Future studies, ideally with larger and more diverse samples, are essential to substantiate and potentially extend these initial findings and ensure the robust application of altruism measures in a wider range of contexts.

Despite these limitations, this study had several strengths. First, its novel examination of altruism in relation to vaccination attitudes in a specific geographic and cultural context adds valuable insight to the existing body of knowledge. In addition, the use of two different measures of altruism (IPIP and ECRC scales) enhances our understanding of altruistic tendencies across age groups, particularly in relation to health behaviors, such as vaccination. This dual-measure approach allows for a more discriminating analysis of altruism’s role in public health strategies. Finally, the study’s focus on a region with traditionally lower vaccination rates provides a unique opportunity to explore how altruism-based interventions could potentially improve vaccine uptake, providing a model for similar regions facing vaccine-hesitancy challenges.

## 5. Conclusions

The study began by exploring altruistic behavior and its influence on vaccination attitudes within different age cohorts in South Tyrol, revealing differentiated motivations that underpin altruistic behavior at different stages of life. The results underscore the diversity of factors that motivate altruism in younger and older age groups, which must be examined separately to fully understand the complexity of altruistic behavior.

In the context of vaccination, the application of the ECRC altruism scale appeared to be robust for older participants, providing clear insights into their altruistic motivations. However, for the younger cohort, the relevance and interpretability of the scale pose challenges, suggesting a cautious approach for its use in this population. The evolving landscape of vaccination attitudes, particularly illuminated by the shifts observed since the onset of the pandemic in 2021, suggests a dynamic relationship between altruism and vaccination behavior that warrants further in-depth analysis. In particular, parental altruism, previously thought to be a significant driver of vaccination, appears to be decreasing in influence, highlighting the fluid nature of altruistic motivations in relation to health behaviors. In the findings, the declining role of parental altruism as a driver of vaccination decisions invites speculation about its relationship with perceived risk. It is plausible that in the post-pandemic context, as the immediate threat of COVID-19 appears to recede, parents’ perceived risk of the virus to themselves and their children may diminish. This reduced sense of urgency could lead to a shift in motivations for vaccination, with immediate personal and family safety considerations taking precedence over broader altruistic intentions.

The differential impact of altruistic behavior across age groups calls for targeted communication strategies that address the specific motivational framework of each demographic. For the younger population, emphasizing the collective benefits of vaccination, such as herd immunity, through altruism-centered messaging, may increase vaccine uptake. Conversely, for the older population, a focus on personal safety and the direct benefits of vaccination may prove more effective.

In conclusion, this study not only contributes to the existing literature on altruism and health behaviors but also highlights the importance of tailoring public health messages to the different motivations of different age groups. The nuanced understanding of the role of altruism in vaccination attitudes revealed by this study provides a valuable foundation for designing more effective health communication campaigns aimed at improving vaccination coverage across all segments of the population. Continued research into the evolving dynamics of altruism and vaccination attitudes will be critical to addressing public health challenges in an increasingly complex world.

## Figures and Tables

**Figure 1 vaccines-12-00454-f001:**
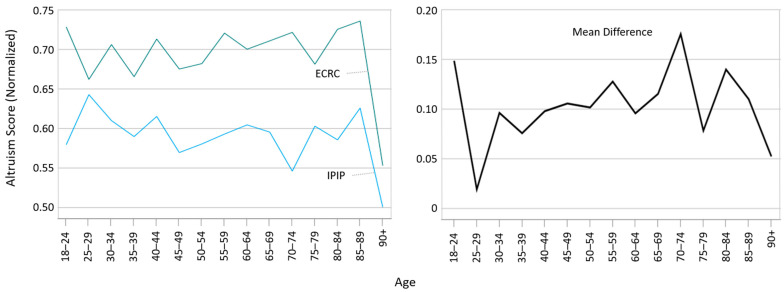
Mean normalized IPIP and ECRC altruism scores (**left** panel) and mean difference of ECRC–IPIP normalized scores (**right** panel) per 5-year age groups.

**Figure 2 vaccines-12-00454-f002:**
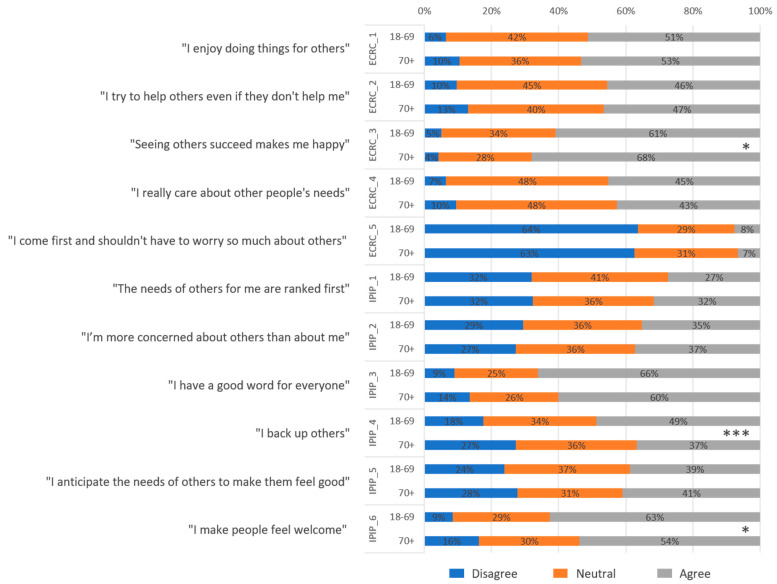
Age-related differences in agreement with altruism statements of ECRC and IPIP scales. For the 6-point Likert responses of the ECRC altruism scale, ‘strongly disagree’ and ‘disagree’ were grouped into a consolidated “Disagree” category, while ‘agree’ and ‘strongly agree’ were combined into an “Agree” category. The mid-range responses, ‘slightly disagree’ and ‘slightly agree’, were unified under the label “Neutral”. Similarly, for the IPIP’s 5-point scale, the ‘strongly disagree’ and ‘disagree’ responses were categorized as “Disagree”, and ‘agree’ and ‘strongly agree’ were labeled as “Agree”. The neutral midpoint remained as “Neutral”. Mann–Whitney U tests, * *p* < 0.05, *** *p* < 0.001.

**Figure 3 vaccines-12-00454-f003:**
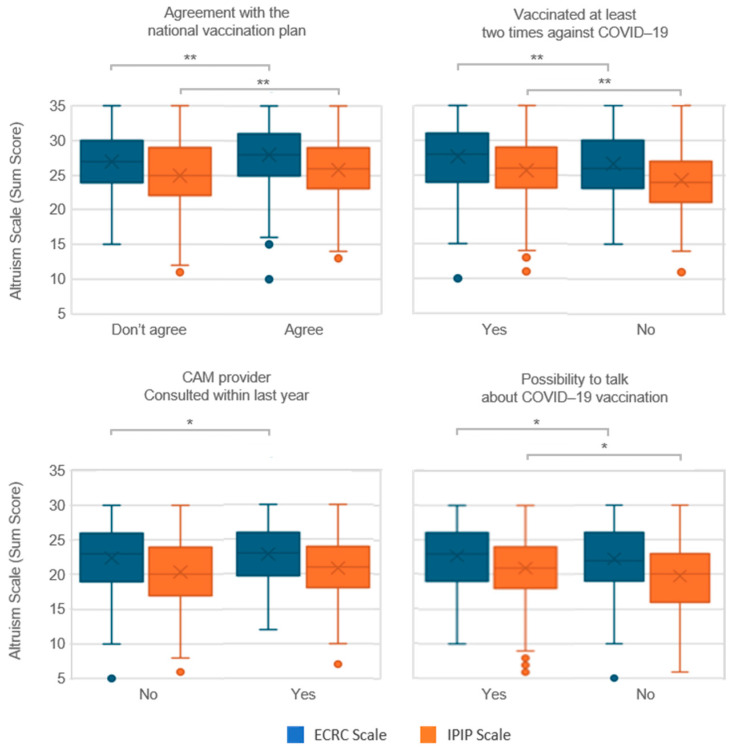
Impact of health behaviors and communication on altruism scores by scale type for the age group 18–69. Mann–Whitney U tests, * *p* < 0.05, ** *p* < 0.01.

**Table 1 vaccines-12-00454-t001:** Demographic and health characteristics, vaccine attitudes, and personal beliefs by age group.

Characteristic	Age Group (%)		*p*-Value ^1^
	18–69	70+	
Female sex	50.7	52.7	n.s.
Italian nationality	88.7	99.0	<0.001
Chronic disease	13.9	42.6	<0.001
Living in urban setting	38.0	55.1	<0.001
Living with children aged 0–17	17.1	0.5	<0.001
Living with COVID-19 patients at risk	10.8	33.7	<0.001
Economic situation			0.017
Better	5.7	1.0	
The same	66.0	71.9	
Worse	25.2	25.6	
Don’t know	3.1	1.5	
Educational level			<0.001
Primary school	12.3	51.2	
Vocational school	29.2	25.9	
High school	33.9	17.1	
University	24.5	5.9	
Vaccination attitudes			
COVID-19 vaccine uptake	88.4	95.6	0.002
Agreement with the national vaccination plan	62.8	73.5	0.003
Flu vaccination in the actual season	12.4	54.6	<0.001
CAM-use within the last 12 months	23.1	10.7	<0.001
Possibility to talk about COVID-19 vaccination	62.7	63.7	n.s.
Childhood vaccination guarantees heard immunity	63.3	70.6	0.046

^1^ Chi-square test. Abbreviations: CAM, complementary and alternative medicine; COVID-19, coronavirus disease 2019; n.s., not significant.

**Table 2 vaccines-12-00454-t002:** Associations between ECRC and IPIP altruism sum scale scores and sample characteristics by age group.

Scales and Variable	ECRC Sum Score Age Group		IPIP Sum Score Age Group	
	18–69	70+	18–69	70+
ECRC ^1^	−	−	0.571 ***	0.559 ***
IPIP ^1^	0.571 ***	0.559 ***	−	−
Age ^1^	n.s.	n.s.	n.s.	n.s.
Female gender ^2^	0.181 ***	0.109 *	0.160 ***	0.184 **
Italian nationality ^2^	n.s.	n.s.	0.061 **	n.s.
Chronic disease ^2^	0.067 **	n.s.	n.s.	n.s.
Urban residency ^2^	n.s.	n.s.	−0.084 ***	n.s.
Living with children aged 0–17 ^2^	n.s.	n.s.	n.s.	n.s.
Living with COVID-19 patients at risk ^2^	n.s.	n.s.	0.055 *	n.s.
Economic situation ^2^	n.s.	n.s.	n.s.	n.s.
Educational level ^2^	0.062 **	n.s.	n.s.	n.s.
Trust in institutions ^2^	0.094 **	0.212 ***	0.195 ***	0.205 ***
Trust in the national vaccination plan ^2^	0.086 ***	n.s.	0.070 ***	n.s.
COVID-19 vaccine uptake ^2^	0.071 **	n.s.	0.064 **	n.s.
Conspiracy thinking ^1^	0.100 ***	n.s.	n.s.	n.s.
CAM use ^2^	0.047 *	n.s.	n.s.	n.s.
Flu vaccination ^2^	0.091 ***	n.s.	n.s.	n.s.
Possibility to talk about COVID-19 vaccination ^2^	0.046 *	0.113 *	0.083 ***	n.s.
Spirituality ^1^	0.307 ***	0.359 ***	0.242 ***	0.307 ***
Well-being ^1^	0.073 **	n.s.	n.s.	n.s.
Agree with importance of heard immunity ^2^	0.110 ***	n.s.	0.074 **	n.s.

^1^ Spearman’s rank correlation. ^2^ Kendall’s tau-b. *** *p* < 0.001, ** *p* < 0.01, * *p* < 0.05, n.s. *p* ≥ 0.05. Abbreviations: CAM, complementary and alternative medicine; COVID-19, coronavirus disease 2019; ECRC, Elderly Care Research Center; IPIP, International Personality Item Pool.

**Table 3 vaccines-12-00454-t003:** Correlations between altruism scales and attitudes towards vaccination policy and trust by age group.

Category	Question	ECRC Scale ^1^ Age Group		IPIP Scale ^1^ Age Group	
		18–69	70+	18–69	70+
Political decision making	I think that decisions about vaccination against COVID-19 made by the public authorities are right	0.079 **	n.s.	n.s.	n.s.
	I think that decisions about mandatory vaccination (not COVID-19) made by the public authorities are right	0.095 **	n.s.	n.s.	n.s.
Trust in COVID-19 vaccination	I believe the vaccination can contain the spread of the virus ^1^	0.141 ***	n.s.	0.084 ***	n.s.
	When all the others are vaccinated against the virus, I don’t need to get vaccinated	n.s.	n.s.	n.s.	n.s.
COVID-19 vaccination is not necessary, because…	…it is not effective	n.s.	n.s.	n.s.	n.s.
	…natural herd immunity is achieved with virus spread and that is quite sufficient	n.s.	n.s.	n.s.	n.s.
	…this disease does not exist/is a normal flu	n.s.	n.s.	n.s.	n.s.
	…the whole thing is only a profit for the pharmaceutical industry	n.s.	n.s.	n.s.	n.s.
COVID-19 vaccination is harmful, because…	...long-term risks are not known	n.s.	n.s.	n.s.	n.s.
	…new vaccines pose additional risks in the mRNA	n.s.	-0.159 *	n.s.	n.s.
	...there are doctors who advise against it	n.s.	n.s.	0.056 *	n.s.
	…a compulsory corona vaccination with prioritization of certain groups will lead to major socio-political discussions	0.086 **	n.s.	n.s.	n.s.
Mandatory vaccination (of children) is unnecessary, because….	…it is not effective	−0.131 ***	n.s.	n.s.	n.s.
	…the natural immune system is enough	−0.112 ***	−0.201 **	n.s.	−0.138 *
	…these diseases to do no longer exist	−0.113 ***	n.s.	n.s.	n.s.
	…the whole thing is only a profit for pharmaceutical industry	−0.111 ***	n.s.	n.s.	n.s.
Mandatory vaccination (of children) is harmful, because….	…the risk is greater than the protection	−0.074 *	n.s.	n.s.	n.s.
	…the vaccines are not controlled enough	−0.072 *	−0.195 **	n.s.	n.s.
	…there are doctors who advice against it	−0.057 *	n.s.	n.s.	n.s.
	…there have been negative vaccine experiences in my family	−0.101 ***	−0.151 *	n.s.	n.s.
What do you actually (in the light of the COVID-19 discussion) think about the situation of the mandatory childhood vaccination?	It is important that the children get the necessary protection	0.215 ***	0.197 **	0.134 ***	0.139 *
	It is important to guarantee heard immunity	0.180 ***	0.140 *	0.128 ***	0.150 *
	I’m worried about the decline of the obligatory vaccination due to the pandemic	0.153 ***	n.s.	0.091 **	n.s.

^1^ Spearman’s rank correlation *** *p* < 0.001, ** *p* < 0.01, * *p* < 0.05, n.s. *p* ≥ 0.05. Abbreviations: COVID-19, coronavirus disease 2019; ECRC, Elderly Care Research Center; IPIP, International Personality Item Pool; mRNA, messenger ribonucleic acid.

**Table 4 vaccines-12-00454-t004:** Predictors of altruism in multivariate linear regression analyses for age groups 18–69 years and 70+ years.

Category	Variable	ECRC Sum Score				IPIP Sum Score			
		18–69 Years N = 1168		70+ Years N = 201		18–69 Years N = 1168		70+ Years N = 201	
		Beta Coefficient	[95% CI]	Beta Coefficient	[95% CI]	Beta Coefficient	[95% CI]	Beta Coefficient	[95% CI]
Constant term		12.625 ***	[11.223;14.027]	11.068 ***	[7.234;14.901]	10.253 ***	[8.498;12.007]	7.252 **	[2.822;11.683]
Demographic factors	Female gender	1.556 ***	[1.065;2.048]	n.s.		1.429 ***	[0.869;1.988]	2.244 **	[0.721;3.767]
	Italian citizenship					1.556 ***	[0.686;2.425]		
	Suffering from a chronic disease	n.s.							
	Urban residency					−1.118 ***	[−0.551;0.982]		
	Living with COVID-19 patients at risk					n.s.			
	Educational level	n.s.							
Vaccination and CAM attitudes	Talk about COVID-19 vaccination	n.s.		n.s.		0.594 *	[0.024;1.164]		
	Agreement with the national vaccination plan	n.s.				n.s.			
	Flu vaccination in the actual season	n.s.							
	Vaccinated at least two times against COVID-19	n.s.				n.s.			
	Agreement with importance of heard immunity	0.594 *	[0.067;1.121]			0.627 *	[0.030;1.224]		
Personal attitudes	Conspiracy thinking	n.s.							
	Trust in institutions	0.077 ***	[0.049;0.106]	0.089 *	[0.014;0.185]	0.047 **	[0.014;0.079]	0.117 **	[0.031;0.204]
	Spirituality	0.342 ***	[0.281;0.403]	0.503 ***	[0.283;0.644]	0.333 ***	[0.263;0.403]	0.412 ***	[0.204;0.620]
	Well-being	n.s.							
	CAM-use	n.s.							
R^2		0.171		0.148		0.139		0.146	

*** *p* < 0.001, ** *p* < 0.01, * *p* < 0.05, n.s. *p* ≥ 0.05. Abbreviations: CI, confidence interval; CAM, complementary and alternative medicine; COVID-19, coronavirus disease 2019; ECRC, Elderly Care Research Center; IPIP, International Personality Item Pool.

## Data Availability

The data presented in this study are available upon request from the corresponding author. The data were not publicly available because the survey was conducted by the statistical office of the regional administrative authority and had politically sensitive content.

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
