# Peer review of "Evolving Altruistic Attitudes towards Vaccination Post COVID-19 Pandemic: A Comparative Analysis across Age Groups"

_vaccines, 2024, doi:10.3390/vaccines12050454_

Round 1
Reviewer 1 Report
Comments and Suggestions for Authors
The paper is generally well written and the study design seems reasonable. The results section seems very long and repetitious. Data that is clearly shown in tables is then repeated in text which usually is not necessary. The paper is also challenged by trying to do too many things at once. It looks at changes in attitudes across time in different age cohorts and at the same time tries to elucidate which tool to measure altruism is more appropriate in each age group. It seems that the size of this study is too small to effectively make statements about age cohort appropriate tools in any setting outside of the study setting, severely challenging the generalizability of the statements that are made. This should be directly addressed in the limitations section of the paper and is a major limitation of the work.
Reviewer 2 Report
Comments and Suggestions for Authors
Thank you for allowing me to review the manuscript entitled "Evolving Altruistic Attitudes Towards Vaccination Post-CPVOD-19 Pandemic: A Comparative Analysis Across Age Groups." The manuscript was well-written and had interesting information. There were a few issues/questions that I did have.
1. The authors state that the study was looking at different age demographics and the efficacy of using different scales to assess altruism across age groups, but actually it was really only two age groups: 18 – 69 years and 70 and over. It seems that the 18-69 group is very broad; people aged 18 and people aged 69 are at very different points in their lives and have different experiences and circumstances, so might it be beneficial to split this into 2-3 age groups (perhaps 18-35, 36 – 49, and 50-69, or something like that)? I understand that this might not be feasible for this study, now, but it might be something to consider for future studies to assess better the utility of the scales for interpreting certain age groups.
2. Based on #1, I would not say that this is "across different age demographics" but rather comparing two age groups – 70 and older vs 18 to 69 years.
3. In the results, second paragraph, they give Cronbach's alpha for ECRC and IPIP for the two age groups for altruism, however, they do not state what the spirituality, well-being, conspiracy thinking, and trust in institution scores are for (ECRC or IPIP?).
Reviewer 3 Report
Comments and Suggestions for Authors
Thank you for the opportunity to review this manuscript on altruistic attitudes to vaccination in the post-pandemic era, with a focus on comparisons between age groups. The study found that the promotion of altruistic behaviour to increase vaccine uptake was more effective in younger adults compared with older age groups. In contrast, emphasising personal safety appears to be more effective when targeting older people. Below are my questions, comments and recommendations:
1) Statistics: Please add a statement on whether α adjustment was done.
2) Interestingly, altruism in general seems to increase with age. And still in the SARS-CoV-2 pandemic, age was positively correlated with enhanced prosocial behaviour. However, this survey focusing on the post-pandemic period found the opposite when it came to vaccination. How do you explain this finding?
3) You stated that fostering trust in institutions has emerged as an age-transcendent strategy to enhance vaccination willingness. During and in the aftermath of the SARS-CoV-2 pandemic, anti-vaccination groups succeeded in damaging this trust. This was outside the scope of the manuscript, but I think it is important to discuss this point. How strong is the influence of anti-vaccination statements in the media and social media?
4) This study found that parental altruism, previously thought to be an important driver of vaccination, appears to be decreasing in influence. Do you think that this might be related to a reduced perception of risk?
5) Do you intend to test age-specfic interventions in relation to vaccination campaigns (e.g. for younger people emphasising the collective benefits of vaccination, for older people focusing on personal safety and the direct benefits of vaccination)? How could such a test be designed?
6) The statements about written informed consent in the Institutional Review Board Statement and the Informed Consent Statement seem to contradict each other. Please check!
Comments on the Quality of English LanguageThe manuscript requires minor editing of English language.
Reviewer 4 Report
Comments and Suggestions for Authors
Overall:
- The study addresses a highly relevant and timely issue, given the ongoing discussions around vaccine uptake and the role of altruism during the COVID-19 pandemic.
- The focus on the evolution of altruism in the post-pandemic era and its impact on vaccination across different age groups presents an original angle that contributes to the existing literature.
- The paper is well-written but review it for minor grammatical issues
Abstract:
- The abstract is generally clear and well-structured, presenting the background, methodology, results, and conclusions in a logical sequence.
- Consider specifying the type of study (e.g., cross-sectional, longitudinal) early in the abstract to give readers a clearer understanding of the research design.
- The transition from discussing the background to detailing the study's methodology and findings is smooth, but there could be a more explicit mention of the study's primary objective to strengthen coherence.
- Introduction:
- The introduction is clear and provides sufficient details
Methods:
The methodology is briefly but effectively described, mentioning the use of a representative sample and specific scales for measuring altruism.
Please consider making adjustments to the regression.
Results:
the results are well presented with explicative tables. However, in tables, please ensure each acronym is specified.
Discussion
the discussion is weel written. Results are cleary summarized in put in relation with previous evidence.
Strengths and limitations are addressed and commented.
Conclusions are drawn based on results
Comments on the Quality of English LanguageThe paper is well-written but review it for minor grammatical issues
